# Immune Checkpoint Expression on Immune Cells of HNSCC Patients and Modulation by Chemo- and Immunotherapy

**DOI:** 10.3390/ijms21155181

**Published:** 2020-07-22

**Authors:** Lisa K. Puntigam, Sandra S. Jeske, Marlies Götz, Jochen Greiner, Simon Laban, Marie-Nicole Theodoraki, Johannes Doescher, Stephanie E. Weissinger, Cornelia Brunner, Thomas K. Hoffmann, Patrick J. Schuler

**Affiliations:** 1Department of Otorhinolaryngology, University Hospital Ulm, Frauensteige 12, 89075 Ulm, Germany; lisa.puntigam@uni-ulm.de (L.K.P.); sandrajeske001@gmail.com (S.S.J.); simon.laban@uniklinik-ulm.de (S.L.); marie-nicole.theodoraki@uniklinik-ulm.de (M.-N.T.); johannes.doescher@uniklinik-ulm.de (J.D.); cornelia.brunner@uniklinik-ulm.de (C.B.); t.hoffmann@uniklinik-ulm.de (T.K.H.); 2Department of Internal Medicine III, University of Ulm, Helmholtzstr. 10, 89081 Ulm, Germany; marlies.goetz@gmail.com (M.G.); greiner@diak-stuttgart.de (J.G.); 3Department of Internal Medicine, Diakonie Hospital Stuttgart, 70176 Stuttgart, Germany; 4Department of Pathology, University of Ulm, Albert-Einstein-Allee 23, 89081 Ulm, Germany; weissinger.stephanie@googlemail.com

**Keywords:** HNSCC, head and neck cancer, immunotherapy, immune checkpoints, checkpoint inhibitors, immunomodulation

## Abstract

Endogenous control mechanisms, including immune checkpoints and immunosuppressive cells, are exploited in the process of tumorigenesis to weaken the anti-tumor immune response. Cancer treatment by chemotherapy or immune checkpoint inhibition can lead to changes of checkpoint expression, which influences therapy success. Peripheral blood lymphocytes (PBL) and tumor-infiltrating lymphocytes (TIL) were isolated from head and neck squamous cell carcinoma (HNSCC) patients (*n* = 23) and compared to healthy donors (*n* = 23). Immune checkpoint expression (programmed cell death ligand 1 (PD-1), tumor necrosis factor receptor (TNFR)-related (GITR), CD137, tumor necrosis factor receptor superfamily member 4 (TNFRSF4) (OX40), t-cell immunoglobulin and mucin-domain containing-3 (TIM3), B- and T-lymphocyte attenuator (BTLA), lymphocyte-activation gene 3 (LAG3)) was determined on immune cells by flow cytometry. PD-L1 expression was detected on tumor tissue by immunohistochemistry. Immune cells were treated with immuno- and chemotherapeutics to investigate treatment-specific change in immune checkpoint expression, in vitro. Specific changes of immune checkpoint expression were identified on PBL and TIL of HNSCC patients compared to healthy donors. Various chemotherapeutics acted differently on the expression of immune checkpoints. Changes of checkpoint expression were significantly less pronounced on regulatory T cells compared to other lymphocyte populations. Nivolumab treatment significantly reduced the receptor PD-1 on all analyzed T cell populations, in vitro. The specific immune checkpoint expression patterns in HNSCC patients and the investigated effects of immunomodulatory agents may improve the development and efficacy of targeted immunotherapy.

## 1. Introduction

### 1.1. Tumor Microenvironment and Squamous Cell Carcinoma of the Head and Neck

Immune evasion is one of the major mechanisms for uncontrolled proliferation and migration of tumor cells [1]. Although the immune system is capable of recognizing and eliminating early malignant cells, tumors evolve to evade immune attack. One of the recently most discussed immune evasion mechanisms is the establishment of an immunosuppressive microenvironment by the tumor [2]. This includes the recruitment of immunosuppressive cells like regulatory T cells (Treg) to the tumor site, as well as the enrichment of these cells in the peripheral blood of tumor patients [3]. Additionally, immune responses are regulated by a number of immunological checkpoints, which promote protective immunity and maintain tolerance [2]. Therefore, immune checkpoints appear to play an important role in the tumor microenvironment and can be manipulated as a mechanism of tumor immune evasion [4]. Head and neck squamous cell carcinoma (HNSCC) is an example of an immunosuppressive disease with lower absolute lymphocyte counts than those found in healthy subjects [5], impaired natural killer-cell activity, and poor antigen-presentation function [6]. Additionally, the frequency and function of tumor infiltrating Treg are elevated in HNSCC [7]. Annually, more than 650,000 cases of head and neck cancer are diagnosed globally [8], frequently in an advanced stage with infiltration of surrounding structures and lymph node metastasis. More than 50% of the patients have recurrence within 3 years [9]. Head and neck cancers are a very heterogeneous group of tumors and the infection with high-risk human papillomaviruses (HPVs) has been implicated in HNSCC pathogenesis. Studies provide strong evidence that the HPV status is an independent prognostic factor for overall survival and progression-free survival, which is consistent with the hypothesis that HPV-related HNSCCs and HPV-unrelated HNSCC differ with respect to the molecular mechanisms underlying their oncogenic process [10,11]. In the event of an advanced HNSCC, a multidisciplinary approach, including surgery, chemotherapy and radiotherapy (RT), is required [12,13]. Patients who experience cancer progression within 3 months after platinum-based chemotherapy of primary or recurrent disease have a median survival rate of 6 months or less [14]. In recent decades, the survival rate has not been significantly increased. Therefore, it is necessary to identify novel biomarkers and potential molecular candidates for targeted cancer therapy. The already mentioned immune checkpoint proteins appear to be a promising therapeutic target for many different cancer types.

### 1.2. Immune Checkpoint Inhibitors in Clinical Use

On the basis of very encouraging results in melanoma patients, inhibitors of the programmed cell death protein 1/programmed cell death-ligand 1(PD1/PD-L1) pathway were also tested in HNSCC patients. Because PD-L1 expression is found in a large subset of HNSCC, PD-1 blockade is a target for immunotherapeutic approaches [15]. The results of the KEYNOTE-048 clinical trial led to the proposed use of the PD-1 inhibitor pembrolizumab plus platinum and 5-fluorouracil (5-FU) as an appropriate first-line treatment for recurrent and metastatic HNSCC and pembrolizumab monotherapy as a first-line treatment for PD-L1-positive recurrent or metastatic HNSCC [16]. These findings suggest that immunotherapy in combination with chemotherapy or as monotherapy is able to achieve an improvement in overall survival of HNSCC patients, but not as superior as initially supposed. In addition to the PD-1/PD-L1 pathway, there are many other immune checkpoints revealing a co-inhibitory or co-stimulatory effect on the anti-tumor immunity [17]. Because of the manifold tumor evasion strategies and different response rates for treatments, combinational treatments are crucial. Therefore, it is essential to know which receptors and proteins are potential targets to be able to develop more precise and effective immunotherapeutic treatments. The present study focuses on the co-inhibitory checkpoints PD-1, t-cell immunoglobulin and mucin-domain containing-3 (TIM3), B- and T-lymphocyte attenuator (BTLA) and lymphocyte-activation gene 3 (LAG3), and the co-stimulatory checkpoints glucocorticoid-induced tumor necrosis factor receptor (TNFR)-related (GITR), CD137, and tumor necrosis factor receptor superfamily member 4 (OX40, TNFRSF4). Meanwhile, several combination immunotherapy trials are ongoing, for example, a combination of a PD-L1 inhibitor (durvalumab) and an agonist of the co-stimulatory immune checkpoint OX40 (NCT02221960), with the aim to achieve a synergistic anti-tumor immune reaction in recurrent or metastatic solid tumors. To investigate whether these combination immunotherapies could also be beneficial for HNSCC patients, it is important to reveal which immune checkpoint receptors are expressed on lymphocytes.

The present study determines the expression of various checkpoints on immune cell subsets in HNSCC patients and their modulation by immuno- and chemotherapy.

## 2. Results

### 2.1. Cohort Characteristics and PD-L1 Status of HNSCC Patients

Peripheral blood samples were obtained from 23 healthy donors (mean age 56 years) and 23 HNSCC patients (mean age 59 years) and fresh tumor tissue was collected from 12 tumor patients (mean age 67 years). Tumor size in the HNSCC group varied (T1–T4) and 16 of the patients already had involved lymph nodes but no distant metastasis (Table 1). The PD-L1 status of the obtained tumor tissue was determined by immunohistochemistry (IHC). A negative PD-L1 status was detected for three patients, 1% positive staining for five patients and 5% positive staining for four of the analyzed patients. Positive staining was additionally quantified using the quickscore method [18]. Negative staining with a quickscore of 0 was detected for three patients. The quickscore of the remaining nine patients ranged over 1–6 (Table 1). Representative staining results are displayed in Figure 1.

### 2.2. Altered Checkpoint Expression on Peripheral Blood Lymphocytes (PBL) of Healthy Donors and HNSCC Patients

Immune checkpoint expression patterns of healthy donors (*n* = 23) and HNSCC patients (*n* = 23) were compared on peripheral immune cell subsets by flow cytometry. In HNSCC patients, PD-1 expression was significantly increased compared to healthy donors on CD8+ T cells (mean value 9.5 ± 7.8% versus 4.5 ± 2.6%) and Treg (mean value 14.5 ± 4.4% versus 11.3 ± 4.2%) (Figure 2A). The GITR expression level was significantly higher on all analyzed T cell subsets of HNSCC patients compared to healthy donors, with the largest difference for CD4+CD39+ Treg (mean value 36.7 ± 11.1% versus 22.5 ± 11.2%, unpaired T test, *p* < 0.0001; Figure 2B). Peripheral Treg of HNSCC patients also displayed significantly elevated levels of the immune checkpoints CD137 (mean value 0.8 ± 0.8% versus 0.4 ± 0.3% healthy controls), while the expression of OX40 on Treg was unchanged (Figure 2C,D). TIM3 expression on peripheral CD8+ T cells was significantly increased in HNSCC patients (Figure 2E). The expression of checkpoints (PD1, GITR, OX40, CD137, TIM3) was determined on all immune cell subsets (CD4+ T_H_ cells, CD8+ T_c_ cells and CD4+CD39+ Treg). The displayed graphs are representative results.

### 2.3. OX40 Upregulation on Treg of HPV-Positive HNSCC Patients

Within the HNSCC group, seven patients tested positively for a HPV infection. To detect possible differences in checkpoint expression between the HPV-positive (HPV+) and HPV-negative (HPV−) tumor patients, we compared expression levels of both groups. We detected significantly increased OX40 levels on Treg of HPV+ tumor patients (mean value of 5.1 ± 1.5% positive cells) compared to HPV− patients (mean value of 2.3 ± 1.3% positive cells) (Figure 3). The other tested immune checkpoints did not display any significant differences between the two groups.

The influence of other clinicopathological characteristics, like tumor size, nodal status and tumor localization on the immune checkpoint expression, were also evaluated, but no relevant effects were detected.

### 2.4. Immune Checkpoint Expression on T-Cell Subsets of PBL and Tumor-Infiltrating Lymphocytes (TIL) in HNSCC Patients

Furthermore, the immune checkpoint expression of intratumoral lymphocytes was analyzed by flow cytometry and compared with the expression profile of paired peripheral lymphocytes of the same HNSCC patients (*n* = 7). Increased PD-1 and GITR expression was detected on all analyzed intratumoral T cell subsets compared to peripheral T cells (Figure 4A,B). Similarly, OX40 expression was significantly upregulated on all T cell subsets isolated from the tumor sites. The OX40 increase was most pronounced on intratumoral Treg (paired T test, *p* < 0.0001). By contrast, the expression of the immune checkpoint BTLA was significantly reduced on all analyzed intratumoral T cell populations compared to the circulating T cells of the patients (Figure 4D). The expression of the co-stimulatory immune checkpoint CD137 was significantly increased on intratumoral CD8+ T cells and Treg, but not on CD4+ T cells, as compared to peripheral blood (Figure 4E).

### 2.5. Influence of In Vitro Chemotherapy Treatment on Immune Checkpoint Expression

The effect of the commonly used chemotherapeutics cisplatin, methotrexate (MTX) and 5-FU on the expression of immune checkpoints on lymphocytes was analyzed in vitro using flow cytometry. The agents were used at three different concentrations C1, C2, and C3 (exact information in the legend of Figure 5) and added to stimulated cells. Checkpoint expression of chemotherapeutically treated cells was compared to that of stimulated cells. We were able to demonstrate effects which were specific for certain cell populations or for certain drugs.

(I) Population-specific effects (Figure 5A): CD8+ T cells displayed significantly reduced CD137 expression for all chemotherapeutic treatments compared to untreated CD8+ T cells. However, in the Treg population (CD4+CD39+), the effect of the same drugs was much less, and it was only significant for cisplatin. Following cisplatin treatment, CD137 on CD8+ T cells was reduced to 15.5 ± 6.8% positive cells versus 29.8 ± 3.6% positive cells for Treg. (II) Drug-specific effects (Figure 5B): PD-1 expression on CD4+ T cells was downregulated by all three chemotherapeutics (mean value of 23.8 ± 6.2% positive cells after cisplatin, Dunnett’s multiple comparison, *p* < 0.001), whereas LAG3 expression did not change significantly after cisplatin treatment. The greatest reduction of LAG3 expression on CD4+ T cells was detected after MTX treatment (mean value of 56.8 ± 9.0% positive cells under untreated condition versus 34.8 ± 9.4% positive cells after MTX treatment). The expression of checkpoints (PD-1, LAG3, and CD137) was determined on all immune cell subsets (CD4+ T_H_ cells, CD8+ T_c_ cells, and Treg). Displayed graphs are representative results.

### 2.6. PD-1 Reduction on Immune Cells after In Vitro Nivolumab Treatment

To assess the effect of in vitro nivolumab treatment on immune checkpoint expression, immune cells were stimulated (1 µg/mL *Staphylococcus* enterotoxin B (SEB) and 300 IU/mL interleukin (IL)-2) and two different concentrations of nivolumab were added to the stimulated cells. After 3 days, cells were harvested and checkpoint expression was determined. The expression of only stimulated cells was compared to nivolumab-treated cells. A significant reduction of surface PD-1 was measured on all immune cell subsets by flow cytometry after nivolumab treatment (Figure 6A). This effect was significant for all three analyzed T cell subpopulations and both applied nivolumab concentrations (5 µg nivolumab *p* < 0.05; 10 µg nivolumab *p* < 0.001) determined by Dunnett’s multiple comparison test). These results were confirmed by a significant downregulation of PD-1 messenger RNA (mRNA) in nivolumab-treated PBL (relative PD-1 mRNA level reduced to 0.63 ± 0.12 after 5 µg nivolumab and 0.36 ± 0.17 after 10 µg nivolumab treatment) compared to untreated PBL measured by reverse transcription polymerase chain reaction (RT-PCR) (Figure 6B). Representative density plots of PD-1 expression on CD8+ T cells are shown to elucidate the reduction of PD-1 on nivolumab-treated CD8+ T cells (37% PD-1 positive cells) compared to untreated cells (67% PD-1 positive cells) (Figure 6C).

## 3. Discussion

### 3.1. Altered Expression of Co-Inihbitory Molecules on PBL of HNSCC Patients

In the present study, we were able to detect significant differences in immune checkpoint expression profiles on freshly isolated PBL of HNSCC compared to healthy donors. Co-inhibitory, as well as co-stimulatory, immune checkpoints were found to be dysregulated on PBL of HNSCC patients. PD-1 was highly upregulated on all peripheral T cell subsets of cancer patients and TIM3 was also increased on CD8+T_c_ cells as compared to healthy donors. Both, PD-1 and TIM3, exhibited an inhibitory effect on the circulating immune cells, which could be associated with a dampened immune response of the host, thereby promoting tumor growth. By contrast, it has also been discussed that PD-1 expression on CD8+ T cells is a sign of antigen-experienced and potentially tumor-reactive cells [19]. Interestingly, PD-1 expression on the immunosuppressive Treg population has been shown to be important for their maintenance and function [20]. Therefore, this receptor exhibits a dual inhibitory potential. It reduces effector T cell function and at the same time enhances Treg proliferation and function.

### 3.2. Dysregulation of Co-Stimulatory Molecules on PBL of HNSCC Patients

A similar role may be played by the co-stimulatory checkpoint GITR. It has been shown, that GITR has a dual positive effect on the immune response of the host by boosting responder T cell function and simultaneously enhancing resistance to Treg-mediated suppression [21]. Moreover, we and others have previously reported GITR upregulation on all T cell subsets as a sign of cell activation [22]. Additionally, we found increased CD137 expression on circulating Treg of HNSCC patients, which is another marker for enhanced T cell activity and is known to have effects on CD8+ T cell proliferation and survival. However, the role of CD137 in Treg function is unclear, because conflicting data suggest that CD137 activity can either expand or restrict Treg cell activation [23]. A recent preclinical study, using anti-CD137 antibodies, demonstrated the depletion of intratumoral Treg after antibody application in mice [24]. Given the fact that we detected increased CD137 expression on all T cell subsets isolated from HNSCC tissue, a targeted treatment with antibodies against CD137 could be conceivable. On the one hand, this could activate CD8+ T cells and on the other hand, it could deplete immunosuppressive Treg and, therefore, strengthen the anti-tumor immunity.

### 3.3. Modulated Immune Checkpoint Expression on Tumor-Infiltrating Immune Cells

In our experiments, TIL of HNSCC exhibited specific differences in checkpoint expression compared to PBL of the same tumor patients. All T cell subsets from the tumor displayed higher PD-1 expression than in PBL. The fact that lymphocytes in the direct tumor microenvironment express high amounts of the inhibitory protein PD-1 indicates exhaustion and lack of functionality. They may, therefore, promote tumor growth in an indirect manner [23]. Recent literature also reported that high PD-1 expression on TIL could also indicate antigen-specific cells [19]. The elevated PD-1 expression on TIL compared to PBL may indicate that these immune cells are more specialized and highly tumor-reactive because of the direct exposure to tumor tissue in the respective microenvironment. Additionally, the upregulation of both co-stimulatory checkpoints GITR and OX40 on all intratumoral T cell subsets is a sign of inflammation within the tumor microenvironment and activation of the surrounding lymphocytes.

Similar to GITR, OX40 exhibits a dual positive effect on the anti-tumor immune response by supporting the survival and expansion of activated T cells and at the same time deactivating the Treg population within tumors, which can further sustain effector T cell function [23]. Because of that, the OX40 upregulation on Treg of HPV+ patients could enhance their anti-tumor immunity through the inhibition of Treg. A phase I clinical trial using an anti-GITR agonist (TRX518) demonstrated reduced circulating and intratumoral Treg cells after antibody application (NCT01239134) but no substantial clinical responses [25]. This led to the initiation of a clinical trial investigating the effects of the anti-GITR antibody in combination with PD-1 blockade to overcome resistance of advanced tumors to anti-GITR monotherapy through reinvigoration of effector T cells (NCT02628574). Additionally, anti-OX40 antibodies are already used in clinical trials. In a neoadjuvant setting, results from a phase Ib clinical trial demonstrated that the GITR antibody MEDI6469 induced T cell proliferation and activity in the tumor microenvironment in 17 treatment-naïve patients with resectable HNSCC (stages III to IVA) [26]. In this context, treatment of HNSCC patients with antibody agonists of GITR or OX40 could provide a powerful tool to diminish the suppressive potential of intratumoral Treg and enhance anti-tumor immunity.

Additionally, we found the co-inhibitory checkpoint BTLA to be significantly downregulated on all T cell subsets isolated from the tumor. BTLA is known to be constitutively expressed on the majority of lymphocytes during their activation phase [27]. However, during their differentiation from activated to resting or memory T cells, BTLA expression is downregulated. The function of these BTLA_low_ intratumoral T cells remains unclear. On the one hand, the downregulation of an inhibitory protein and differentiation processes of T cells could provide more potent antitumor effector cell properties [28]. On the other hand, because these cells remain in a more differentiated state, their proliferative potential and cytokine expression profile is restricted. This topic needs further investigation to determine whether BTLA could be a potential target in HNSCC immunotherapy.

### 3.4. Chemo-Immunomodulation of Circulating Immune Cell Subsets

In addition to these findings, the present study revealed diverse influence of chemotherapeutical treatments on the immune checkpoint expression on lymphocytes, in vitro. One major observation was the reduced responsiveness of in vitro chemotherapeutically treated Tregs in comparison to other subpopulations regarding immune checkpoint reduction. Recent studies revealed that Treg persist in the circulation of cancer patients after adjuvant chemoradiotherapy (CRT), whereas the frequency of other lymphocyte populations is reduced [29]. This implies that Treg are resistant to CRT, but the underlying mechanisms remain unclear. In our experiments, after the application of platinum-based chemotherapy, a significant change in immune checkpoint expression was detected in Treg. This resistance may result from low Treg proliferation rates or enhanced capability for DNA repair [30]. Furthermore, we detected drug-specific effects on immune checkpoint expression, possibly depending on the mode of action of the specific chemotherapeutic drug. Both findings need further analysis, because immunotherapy is frequently applied after previous chemotherapeutic treatment regimens. When the in vivo application of chemotherapeutics has similar effects on immune checkpoint expression of lymphocytes, subsequent immunotherapies possibly need to be adjusted to effectively enhance anti-tumor immunity. A retrospective clinical study demonstrated that colorectal cancer patients strongly benefited from a combined chemo-immunotherapy regimen (GOLFIG) with respect to progression-free survival and overall survival. In particular, chemotherapeutically pre-treated patients displayed increased anti-tumor responses [31]. Several recent trials suggest that the antigen load is critical for an efficient treatment response and that immune-modulating treatments, including chemotherapy and RT, could increase the chance to benefit from checkpoint inhibitors and immunotherapy [32]. Additionally, in the field of head and neck cancer, randomized combinatorial studies are in progress, comparing beneficial effects of RT-cisplatin with and without subsequent nivolumab treatment (NCT03576417). Not only chemo- and radiotherapy are conceivable combinatorial treatments to immunotherapy but also a monoclonal antibody against the vascular endothelial growth factor (VEGF) called bevacizumab has shown immunomodulatory effects in advanced squamous non-small lung cancer patients [33]. The mPEBv regimen (metronomic chemotherapy regimen with dose-fractioned cisplatin and oral etoposide (mPE) +/− bevacizumab) used in the respective study induced an anti-tumor immunization that could be enhanced by subsequent PD-1/PD-L1 checkpoint inhibitors [33].

### 3.5. PD-1 Reduction after In Vitro Checkpoint Blockade

The influence of chemotherapy on immune checkpoint expression is not the only aspect that is currently poorly understood. Additionally, the effects of PD-1-antibody mediated anti-tumor response and the respective downstream signaling have not been investigated in depth. In the present study, we analyzed possible effects of in vitro PD-1 blockade with nivolumab on the immune checkpoint expression of T cell subpopulations and detected PD-1 downregulation on all analyzed subsets by flow cytometry and RT-PCR. During the course of nivolumab treatment of cancer patients, the downregulation of the co-inhibitory immune checkpoint could exhibit an additional positive effect on the anti-tumor response. In addition to this, the downregulation of a constantly blocked receptor may be a protective mechanism for the development of resistance to the PD-1 antibody. The relevance of the PD-1/PD-L1 signaling pathway in HNSCC has already been described because PD-L1 is expressed on a large subset of HNSCC [15]. It is necessary to investigate whether these interesting in vitro results can be reproduced in the in vivo situation to further understand the exact mechanisms underlying the enhanced anti-tumor immunity after PD-1 blockade. Another factor that directly influences the effect of immunotherapy are the human leukocyte antigen (HLA) alleles in tumor patients. A recent study demonstrated a predictive role of germinal class I HLA and DRB1 genotype for treatment with the PD-1 inhibitor nivolumab [34]. The results indicated a poor outcome in patients with non-small cell lung cancer who were negative for the most frequent HLA-A alleles, whereas patients expressing HLA-A*01 displayed prolonged progression-free survival and overall survival [34]. A link between specific HLA haplotypes and the expression of immune checkpoints in HNSCC patients is conceivable [35] and should be investigated in future studies to find reliable biomarkers for effective immunotherapy and to protect patients that do not benefit from these therapies because of severe immune-related adverse events.

## 4. Materials and Methods

### 4.1. Patient Samples and Specimens

Peripheral blood samples were obtained from 23 healthy donors and 23 HNSCC patients, who were treated in the local ENT-department (Table 1). Additionally, tumor tissue was collected from a subgroup of 12 of the HNSCC patients to isolate TIL. All subjects signed the informed consent form approved by the local ethical review committee (#255/14). Tumor patients were treatment-naïve before sample collection but subsequently received different treatment according to their tumor stage. Four patients underwent only primary surgery, eight patients had adjuvant chemoradiotherapy (CRT) after surgery, four patients had primary RCT, and seven patients had an adjuvant RT after surgery.

### 4.2. Isolation of Peripheral Lymphocytes

Blood samples from cancer patients and healthy donors (40–50 mL) were drawn into anticoagulation tubes. Lymphocyte isolation was performed with the Leucosep™ System (Greiner bio-one, Frickenhausen, Germany) according to the instruction manual. Leucosep tubes were prepared with 15 mL Biocoll separation medium (Biochrom, Berlin, Germany) and centrifuged at 1000× *g* for 1 min. Whole blood samples were directly added to the Leucosep tubes and centrifuged at 800× *g* for 15 min with switched-off bakes. Following centrifugation, interphase consisting of peripheral blood lymphocytes was isolated. The isolated cell fraction was washed once with phosphate-buffered saline (PBS) by centrifugation at 400× *g* for 10 min. To avoid contamination with erythrocytes, the lymphocytes were incubated with 10 mL Red Blood Cell Lysis Solution (MACS Miltenyi, Bergisch Gladbach, Germany) and diluted 1:10 with sterile water for 10 min at room temperature. The lysis reaction was stopped by adding 40 mL PBS and a centrifugation step at 400× *g* for 10 min. The supernatant was discarded and lymphocytes were collected in 10 mL PBS and counted with an automated cell counter (BioRad, Hercules, CA, USA).

### 4.3. Isolation of TIL

Tumor tissues were directly obtained from the surgical room in NaCl solution. These were minced into small pieces and digested with collagenase I in Roswell Park Memorial Institute (RPMI) medium without any supplements at 37 °C for 2–3 h. The digested tissue was meshed through a cell strainer with 100 µm pores. The single-cell suspension was underlaid with Biocoll separation medium and centrifuged at 800× *g* for 15 min with switched-off brakes to isolate the lymphocytes.

### 4.4. Stimulation of PBL In Vitro

Lymphocytes were cultured in RPMI medium with 10% fetal bovine serum (FBS, Biochrom, Berlin, Germany) and 1% penicillin/streptomycin (Pan-Biotech, Aidenbach, Germany) and seeded at a concentration of 1 × 10^6^cells/mL. For stimulation, 1 µg/mL SEB (Sigma Aldrich, St. Louis, MI, USA) and 300 IU/mL IL-2 (Immunotools, Friesoythe, Germany) were added to the medium and the lymphocytes were cultured at 37 °C under an atmosphere of 5% CO_2_ for two days. Lymphocytes were harvested, washed with PBS once, and transferred to fluorescence-activated cell sorting (FACS) tubes (3 × 10^6^ cells per staining).

### 4.5. Chemotherapeutic Treatment of PBL

Isolated PBL were cultured in RPMI medium with 10% FBS and 1% penicillin/streptomycin and seeded at a concentration of 1 × 10^6^ cells/mL. All cells were stimulated with SEB and IL-2 (see Section 4.4.). Additionally, the cells were treated with different chemotherapeutics for 3 days, these agents being added at the beginning of the culture. Three different chemotherapeutics were tested in three different concentrations. Cisplatin (TEVA, Petach Tikva, Israel) and MTX (MEDAC, Wedel, Germany) were applied in the concentrations 2 µg/mL (C1), 5 µg/mL (C2), and 8 µg/mL (C3), whereas 5-FU (MEDAC) was applied in the concentrations 10 µg/mL (C1), 25 µg/mL (C2), and 50 µg/mL (C3). The applied concentrations were in the range of those found in cancer patients [30]. A total of 5 × 10^6^ cells were seeded per approach. For the control, some cells only received the stimulation without any chemotherapeutic treatment. After 3 days, cells were harvested and stained for immune checkpoint expression.

### 4.6. Nivolumab Treatment of PBL

Isolated PBL were cultured in RPMI medium with 10% FBS and 1% penicillin/streptomycin and seeded at a concentration of 1 × 10^6^ cells/mL. All cells were stimulated with SEB and IL-2 as described in 4.3.1. Additionally, the cells were treated with two different concentrations (5 µg/mL, 10 µg/mL) of the PD-1 inhibitor nivolumab (Opdivo, Bristol Myers Squibb, New York City, NY, USA) at the beginning of the culture. A total of 5 × 10^6^ cells were required for each treatment. As control, some cells received only the stimulation without nivolumab treatment. Cells were harvested after 3 days and stained for immune checkpoint expression.

### 4.7. Antibodies and Reagents

The following anti-human monoclonal antibodies (mAb) were used for flow cytometry staining: CD8-FITC, CD137-PE, CD39-PE-Cy7, CD4-AF700, LAG3-PE, PD-1-PE, PD-L1-PE (all eBioscience. Waltham, USA), BLTA-BV421, GITR-BV421, OX40-PE-Cy5, TIM3-PB (all Biolegend, London, UK), and CD45-AMCyan (BD Bioscience, San Jose, CA, USA). All antibodies were pre-titrated using stimulated and non-stimulated PBL to determine optimal staining dilutions. For IHC, PD-L1 (E1L3N) XP Rabbit mAb was purchased from Cell Signaling Technology, Danvers, MA, USA.

### 4.8. Flow Cytometry

For surface protein staining, cells were centrifuged at 400× *g* for 10 min in FACS tubes (up to 1 × 10^7^ cells) and the supernatant was discarded. The cell pellet was re-suspended and antibodies were added to the residual supernatant and incubated for 30 min at room temperature in the dark. The immune checkpoint receptor staining was performed with three different panels, containing four subpopulation markers and three checkpoint markers each. Following antibody incubation, the cells were washed once with 2 mL FACS buffer (PBS with 5% bovine serum albumin) and diluted in 250 µL FACS buffer for measurement with a Gallios flow cytometer (Beckman Coulter, Brea, CA, USA).

### 4.9. Gating Strategy

The lymphocyte population was gated according to the characteristic size of the lymphocyte fraction with forward and sideward scatter. Cytotoxic T cells were defined as CD8+ cells. T helper cells were defined as CD4+ cells and Treg as CD4+CD39+ cells according to previous literature that identified this subgroup as a highly immunosuppressive subpopulation [36]. A representative gating strategy figure can be found in the Appendix A.

### 4.10. RT-PCR

Total RNA was extracted from PBLs using the RNeasy Mini Kit (Qiagen, Hilden, Germany). RNA concentrations were measured with the Infinite M200 Pro (Tecan, Männedorf, Switzerland). Quantitative RT-PCR was performed using the QuantiTect Reverse Transcription Kit (Qiagen) according to manufacturer instructions and conducted on the Lightcycler96 (Roche, Basel, Switzerland). Relative quantification of the target gene mRNA expression was calculated with the 2^-∆∆C^_t_ method. The expression level of the target gene was normalized to the level of an endogenous control (RPL13A housekeeping gene). The following specific primers were used: human PD-1, 5’-ggccaggatggttcttagact-3’ (forward) and 5’-gaagctgcaggtgaaggtg-3’ (reverse); human RPL13A, 5’-cccctgtttcaagggataaga-3’ (forward) and 5’-gaccatcaagcaccaggac-3’ (reverse).

### 4.11. IHC

Optimal results were obtained for IHC by pretreating the formaldehyde-fixed paraffin-embedded (FFPE) tissue with Tris/EDTA buffer (pH 9, steamer) and with antibody against PD-L1, clone E1L3N (Cell Signaling) with a dilution of 1:100 at 4 °C overnight. The DAKO REAL Detection System, Alkaline Phosphatase/Red (DAKO, Carpintera, CA, USA) with chromogen red was used according to manufacturer’s protocol and established protocols [37]. As negative control, we performed staining without antibody, while tonsil and placenta served as positive controls.

### 4.12. Statistics

All data are presented as means of at least three experiments ± standard deviation (SD). Data were analyzed for their distribution by the D’Agostino and Pearson normality test. In case of normal distribution, two data sets were compared by paired or unpaired T-test, and more data sets were compared by Dunnett’s multiple comparison test. When data were not normally distributed, data sets were compared by Mann–Whitney test. A false discovery rate approach was used to correct for multiple tested hypothesis (Q = 1%). All statistical tests were performed using GraphPad Prism (GraphPad Software, San Diego, CA, USA). *p*-values < 0.05 were considered to be significant.

## 5. Conclusions

An improved understanding of the immune response to cancer, as well as patient selection and development of suitable biomarkers, is essential to increase the number of patients who benefit from immunotherapies. The present study describes the unique expression pattern of immune checkpoint receptors on lymphocytes of HNSCC patients with the aim of a better understanding of the suppressive tumor microenvironment. These insights could serve as a basis for further investigations in terms of beneficial combinational or mono-immunotherapy for head and neck cancer patients.

## Figures and Tables

**Figure 1 ijms-21-05181-f001:**
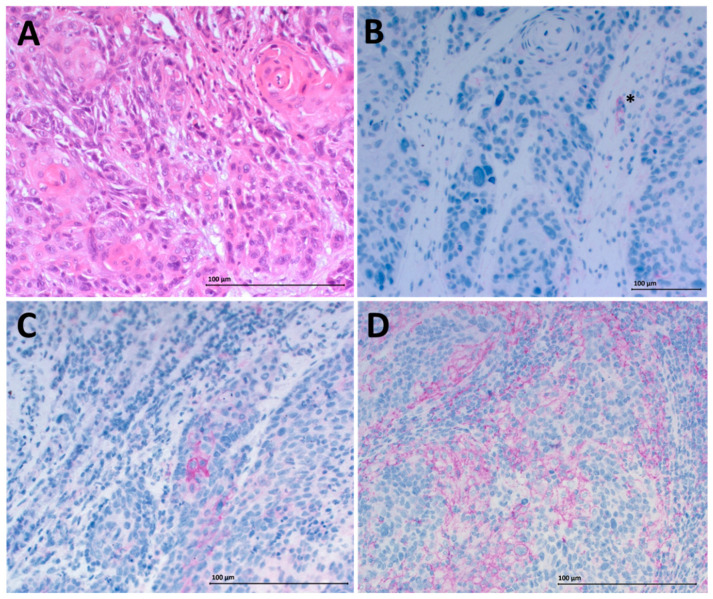
Immunohistochemistry for programmed death ligand 1 (PD-L1) staining. Hematoxylin-eosin-staining of tumor tissue with 20× magnification is shown in (**A**). Negative PD-L1 staining (0% PD-L1 positive tumor cells) is displayed in (**B**). The asterisk (*) marks a positive internal control (immune cell). (**C**) Shows low (1% positive cells) PD-L1 expression on tumor cells and (D) intermediate (5% positive cells) expression. (**B**–**D**) are displayed with 20× magnification. The scale bars represent a length of 100 µm.

**Figure 2 ijms-21-05181-f002:**
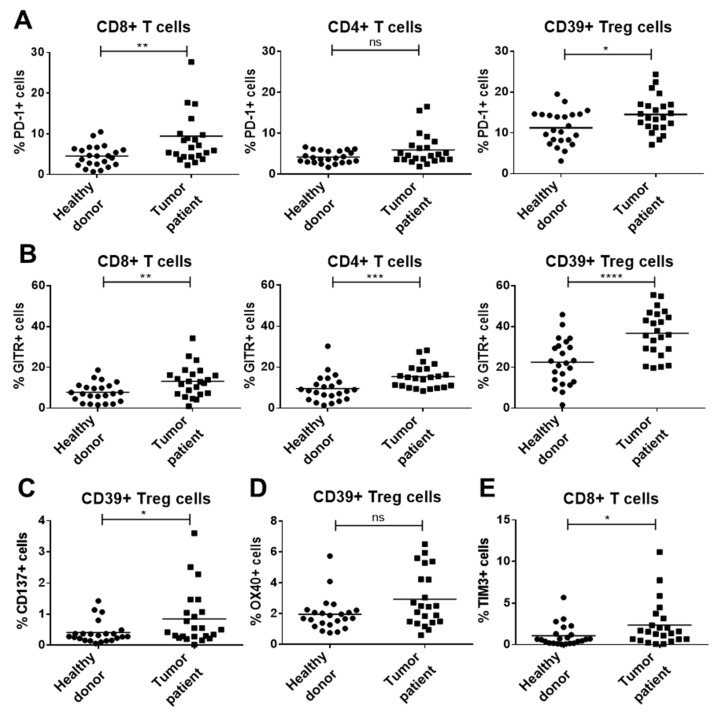
Expression of different immune checkpoints on peripheral blood immune cell subsets was analyzed by flow cytometry. Expression patterns of 23 healthy donors and 23 head and neck squamous cell carcinoma (HNSCC) patients were compared. (**A**) PD-1 expression was significantly increased on CD8+ T cells and regulatory T cells (Treg) from HNSCC patients. (**B**) The expression of glucocorticoid-induced tumor necrosis factor receptor (TNFR)-related (GITR) was significantly higher on all analyzed T cell subsets of HNSCC patients compared to healthy donors. (**C**) Circulating Treg of tumor patients displayed elevated levels of the immune checkpoints CD137. (**D**) Tumor necrosis factor receptor superfamily member 4 (TNFRSF4) (OX40) expression on Treg was not significantly increased. (**E**) t-cell immunoglobulin and mucin-domain containing-3 (TIM3) expression on cytotoxic CD8+ T cells isolated from HNSCC patients was significantly increased. *p*-values < 0.05 were considered to be significant with (*), *p*-values < 0.01 (**), *p*-values < 0.001 (***) and *p*-values < 0.0001 (****), *p* > 0.05 (ns).

**Figure 3 ijms-21-05181-f003:**
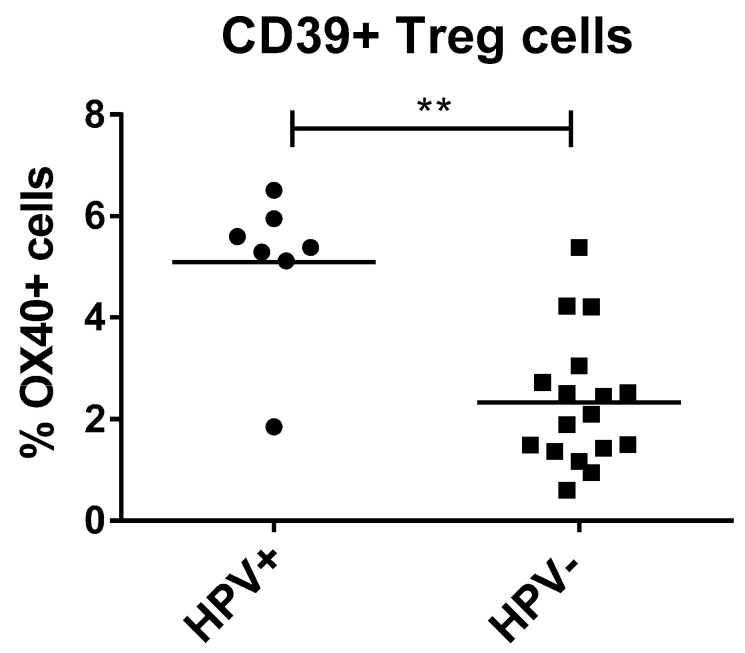
Co-stimulatory immune checkpoint OX40 expression on circulating Treg of human papillomaviruses (HPV)+ tumor patients (*n* = 7) was significantly increased compared to HPV− patients (*n* = 16). Mann–Whitney test was used to determine significance, with *p* = 0.0015. The expression of immune checkpoints on peripheral blood lymphocytes was measured by flow cytometry. *p*-value < 0.01 was considered to be significant with (**).

**Figure 4 ijms-21-05181-f004:**
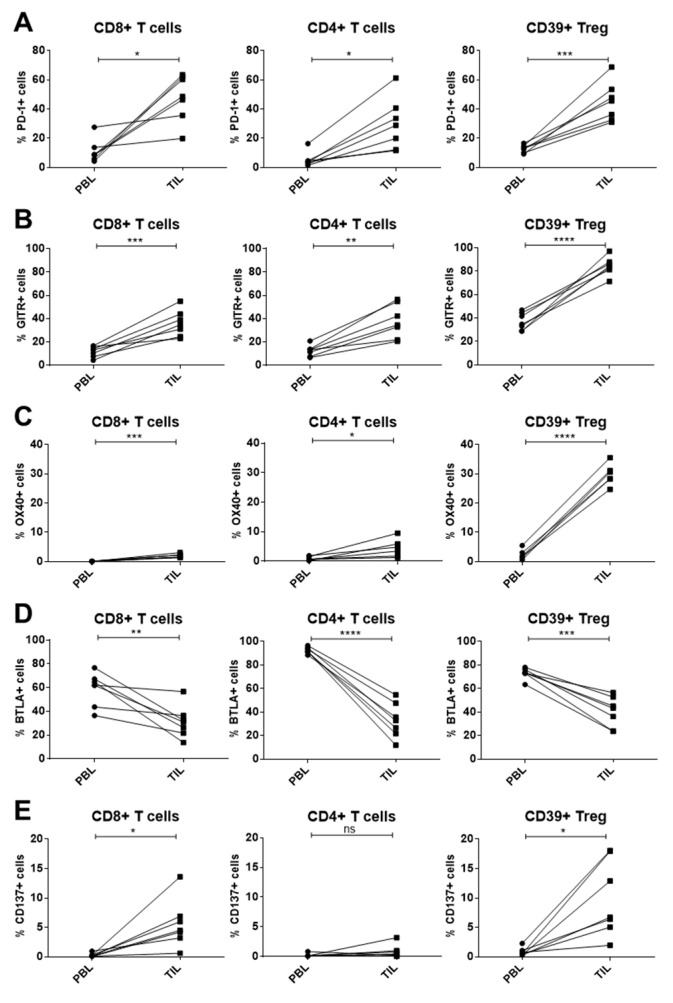
Immune checkpoint expression was compared between paired peripheral blood lymphocyte and tumor infiltrating lymphocyte subsets of the same HNSCC patients (*n* = 7). (**A**) All analyzed T cell subsets isolated from the tumor site revealed significantly higher PD-1 expression compared to circulating T cells. (**B**),(**C**) Similarly, GITR and OX40 expression was significantly upregulated on TIL subsets, both displaying the largest difference on the regulatory T cell (Treg) population (*p* < 0.0001). (**D**) B- and T-lymphocyte attenuator (BTLA) expression on tumor-infiltrating T cells was decreased compared to circulating T cells in all presented subpopulations. (**E**) On CD8+ T cells and CD39+ Treg isolated from the tumor an increase in CD137 expression could be measured, whereas the population of CD4+ T cells did not show a significant difference. *p*-values < 0.05 were considered to be significant with (*), *p*-values < 0.01 (**), *p*-values < 0.001 (***), and *p*-values < 0.0001 (****), *p*-values > 0.05 (ns).

**Figure 5 ijms-21-05181-f005:**
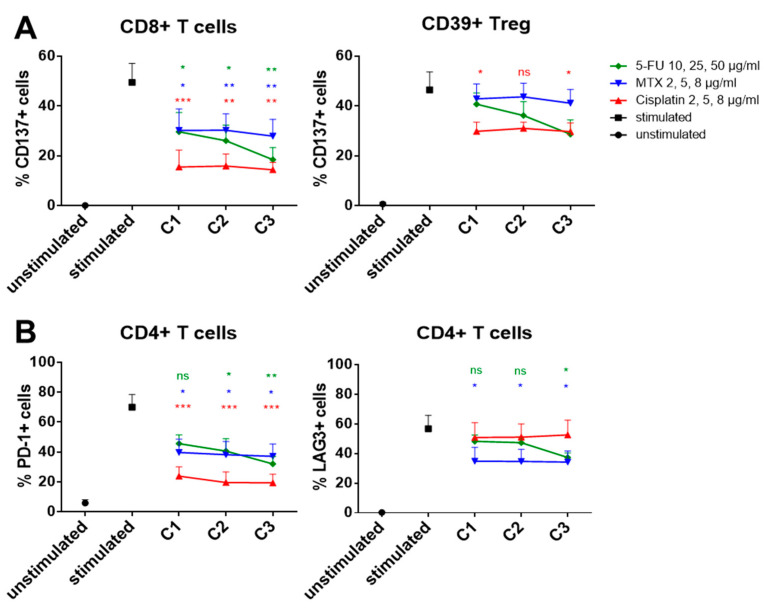
Lymphocytes of five healthy donors were isolated, seeded and stimulated with 1 µg/mL *Staphylococcus* enterotoxin B and 300 IU/mL interleukin-2 (IL-2). Additionally, the cultures were treated with different chemotherapeutics. After 3 days, cells were harvested and immune checkpoint expression was measured by flow cytometry. (**A**) CD8+ T cells displayed significantly reduced CD137 levels after treatment with 5-fluoruracil (5-FU), methotrexate (MTX), and cisplatin. By contrast, the effect of the chemotherapeutics on Treg was significantly impaired regarding CD137 reduction. In (**B**) the different effects of the applied chemotherapeutic on the analyzed immune checkpoints is shown. PD-1 expression on CD4+ T cells was reduced by all agents, but the most by cisplatin treatment. By contrast, MTX had the strongest effect on lymphocyte-activation gene 3 (LAG3) expression on the same cell population. *p*-values < 0.05 were considered to be significant with (*), *p*-values < 0.01 (**), *p*-values < 0.001 (***), *p*-values > 0.05 (ns).

**Figure 6 ijms-21-05181-f006:**
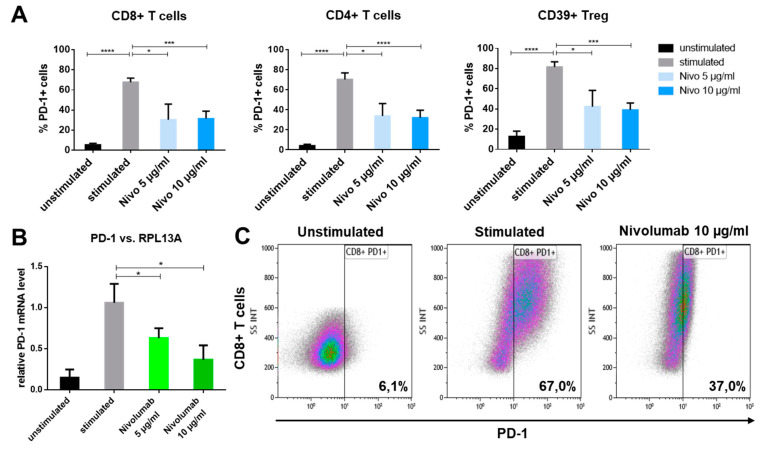
Different concentrations of nivolumab were applied in vitro to lymphocytes for 3 days. (**A**) Subsequent fluorescence-activated cell sorting (FACS) analysis revealed a reduction of PD-1 expression after nivolumab (nivo) treatment on all analyzed T cell subsets (*n* = 5 healthy donors). (**B**) A similar significant reduction was shown on PD-1 mRNA by RT-PCR of nivolumab-treated lymphocytes. RPL13A was used for normalization of mRNA levels (*n* = 3 healthy donors). (**C**) Representative density plots of CD8+ T cells showing the reducing effect of in vitro nivolumab treatment on PD-1 expression. *p*-values < 0.05 were considered to be significant with (*), *p*-values < 0.001 (***) and *p*-values < 0.0001 (****).

**Table 1 ijms-21-05181-t001:** Clinicopathological characteristics of HNSCC patients and the control group.

	Control Group	HNSCC Patients (Blood)	HNSCC Patients (Tumor Tissue)
***n* (female/male)**	23 (13/10)	23 (9/14)	12 (5/7)
**Age (±SD) range (y)**	56 ± 19 (27–84)	59 ± 11 (37–74)	67 ± 9 (49–77)
**Stage (*n*)**			
T1		4	2
T2		8	6
T3		5	2
T4		6	2
**Nodal status (*n*)**			
N0		7	2
N1		5	3
N2		7	7
N3		4	0
**Location (*n*)**			
Oral cavity		5	1
Oropharynx		10	8
Hypopharynx		4	1
Larynx		4	2
**PD-L1 status on tumor tissue (*n*)**			
0%			3
≥1%			5
≥5%			4
**PD-L1 Quickscore**			
0			3
1			3
2			2
4			2
6			2
**HPV status**			
Positive		7	
Negative		16	
**Overall survival at 3 y**		22/23	
**Disease-free survival at 3 y**		21/23

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
