# Peer review of "Immune Checkpoint Expression on Immune Cells of HNSCC Patients and Modulation by Chemo- and Immunotherapy"

_ijms, 2020, doi:10.3390/ijms21155181_

Round 1

Reviewer 1 Report

The manuscript is potentially interesting and describes new mechanisms by which it is possible to increase the sensitivity of cancers to immunological check point inhibitors. The manuscript is well written and the review of the literature was well performed adding all the relevant issues on the specific focus. However, some major pitfalls limit the publication priority of the manuscript that has to be revised and re-submitted by the authors for a second round review.

Major changes:

1) Figure 1. Please show better quality images in which the positivity of the cells to the markers tested is indicated with a brown colour (positive immunohistochemical image).

2) For all the patients enrolled and evaluated in the study it should be interesting to know the treatments performed and the clinical outcome with eventually the DSF and the OS. 

3) It should be nice to know also about the HLA haplotype of the patients enrolled in the study and to correlate the expression of the PDL1 to the haplotype expression.

4) The "Discussion" section of the revised version of the manuscript should be implemented citing some relevant manuscripts on this specific issue. The following manuscripts should be cited for the correlation between the immmunological check point inhibitors and the HLA: PMID: 32554614, PMID: 31865873; on the role of chemotherapy in sensitizing cancer patients to immunotherapy please cite the following recent manuscripts: PMID: 31781481, PMID: 29221287, PMID: 27993089. Please cite also the most relevant results described inside the manuscripts indicated above.

5) Some misreadings throghout the text need correction.

Reviewer 2 Report

English should be revised 

Abstract is confusing due to "was determined on immune  cells by flow cytometry." but results deal also with IHC analysis. Abstract is not informative of the whole experiment workflow. It is confusing

In results when reporting p value, please report (Name test, p value = xxxx); please report also mean+SD when comparing the the level of marker expression.

When reporting results coming from general analysis, it is confusing, please report if you are dealing with IHC or Flow analysis. 

To improve the quality of the study, please perform Spearman or Pearson correlation analysis between markers and clinic-pathological features. 

In IHC please report how did you perform the scoring. I suggest Detre et al. method A "quickscore" method for immunohistochemical semiquantitation: validation for oestrogen receptor in breast carcinomas, although a published meta-analysis on PD-L1 report as pos/neg cut-off 5% positivity in most of studies High PD-L1 Expression in the Tumour Cells Did Not Correlate With Poor Prognosis of Patients Suffering for Oral Squamous Cells Carcinoma: A Meta-Analysis of the Literature

Although the manuscript is of interest, authors performed different experiments and test but they are reporting results in a confusing manner. 

Improve quality of Figure 1

Why did you perform PD-L1 only as IHC?

The manuscript should be re-organized in more paragraphs to improve the understanding of the whole study workflow and analysis

English would be revised

Reviewer 3 Report

It is a well written paper, with a notable immuno-oncological background and a sound experimental basis. Still the design appears heavily biased by not considering a key element of head and neck oncology. In fact, head and neck cancers are extremely heterogeneous [bosch] under a clinical molecular and therefore presumably immunological point of view.  What is more about 20% of HNSCC are virus induced (HPV and EBV related), and this group is even staged separately in the new AJCC TNM and is expected to have clear peculiarities under an immunological point of view. The site of origin in the head and neck cancers under study is listed in a table but apparently not included in the statistical analysis, while the best design would have been considering only one site and only virus-induced or not-induced primaries. HPV status is not even mentioned. I recommend trying to analyze separately the different subsite or at least check for differences and to at least evaluate HPV status in oropharyngeal primaries. The other heterogeneity factor under a clinical point is staging but I understand that with the original study design this bias is very hard to recover or mitigate with further statistical analysis.

Round 2

Reviewer 1 Report

Tge authors have performed sufficient efforts to ameliorate their manuscript

Reviewer 2 Report

the authors satisfied the reviewers requests